# Comparison of ultrasound and dynamic MRI for the measurement of diaphragmatic excursion: A prospective single-center study

Clara Delplancke[1,2☯*], Etienne Charpentier[3☯‡], François Grolleau[4‡], Anne Hernigou[3‡], Hélène Nougué[1,2‡], Françoise Le Pimpec-Barthes[5‡], Bernard Cholley[1,2☯], Matthieu Daniel[1,2☯]

1 Department of Anesthesiology and Intensive Care Medicine, Hôpital Européen Georges Pompidou, AP-HP, Paris, France, 2 Université Paris Cité, INSERM, IThEM, Paris, France, 3 Department of Radiology, Hôpital Européen Georges Pompidou, AP-HP, Paris, France, 4 Université Paris Cité and Université Sorbonne Paris Nord, Inserm, INRAE, Center for Research in Epidemiology and StatisticS (CRESS), and Centre d'Epidémiologie Clinique, Hôpital Hôtel Dieu, AP-HP, Paris, France, 5 Department of Thoracic Surgery, Hôpital Européen Georges Pompidou, AP-HP, Paris, France

☯ These authors contributed equally to this work.
‡ FG, AH, HN and FLP-B also contributed equally to this work.
* clara.delplancke@gmail.com

## Abstract

### Objectives

Ultrasound (US) measurements of diaphragmatic excursion (DE) are widely used to provide a non-invasive assessment of the diaphragmatic function at the bedside, especially in intensive care. However, this measurement has never been validated against a less operator-dependent technique such as MRI. Dynamic MRI is the only imaging modality that creates a four-dimensional reconstruction of the diaphragm. The primary objective of this study was to assess the agreement between DE obtained using dynamic MRI with those obtained using ultrasound. The secondary objectives were to define DE thresholds for the diagnosis of DD using MRI and to compare the performance of US and MRI to diagnose DD.

### Methods

Prospective single-center study in which consecutive outpatients referred for a dynamic thoracic MRI were included. This study was conducted at a university hospital in Paris, where there was daily access to ultrasound (US) and extensive expertise in diaphragmatic MRI The DE of each hemi-diaphragm was measured sequentially using ultrasound and MRI in random order, during spontaneous breathing (SB) and forced inspiration (FI) by independent observers blinded to each other. We analyzed the agreement between DE obtained using US and MRI for each hemi-diaphragm.

### Results

We enrolled forty-five patients, aged 58 ± 36 years, of which twenty-eight (68%) had a confirmed DD. During SB, the mean bias for DE measurement was –3.8 mm, 95% CI [–7.1; –0.6] for the left hemi-diaphragm, and 1.0 mm, 95% CI [–3.5; 5.5] for the right

**Data availability statement:** All relevant data are within the paper.

**Funding:** The author(s) received no specific funding for this work.

**Competing interests:** The authors have declared that no competing interests exist.

hemi-diaphragm. Limits of agreement (millimeters) were [–25; 17] on the left side, and [–28; 30] on the right side. MRI threshold values for DE defining dysfunction were 11 mm for quiet SB, and 38 mm for FI. These thresholds had a sensitivity of 77.7% and a specificity of 77.4% during SB, with an AUC of 0.86.

## Conclusion

US and MRI provide different values for DE, probably because the measurements were not obtained exactly at the same localization. Nevertheless, diagnostic performances of MRI and US to recognize DD appeared comparable.

## Introduction

Diaphragmatic dysfunction (DD) is a common problem in critically ill patients [1–4], and may be responsible for prolonged duration of mechanical ventilation and even failure to wean from the ventilator [5]. This dysfunction results from a neuromuscular disorder that can involve the central nervous system, a peripheral neuropathy, or a myopathy. The incidence of DD varies between 29 to 40% in ICU patients, 20% to 40% following upper-abdominal surgeries, and up to 83% after open thoracic procedures [2,4,6].

Transdiaphragmatic pressure measurements combined with electromyographic recordings under supramaximal electrical or magnetic stimulation of the phrenic nerve represents the gold standard for assessing diaphragm function [7]. But, due to its complexity and invasiveness, this technique cannot be used routinely in critically ill patients. Ultrasonography (US) can provide a simple, rapid and non-invasive assessment of diaphragmatic function at the bedside [1,8–13]. However, this measurement has never been validated against either the gold standard or another robust imaging technique: less operator-dependent [14].

Dynamic MRI is the only imaging modality that creates a four-dimensional reconstruction of the entire movement of the diaphragm and allows visualization of the precise diaphragmatic course using echo-gradient and FIESTA sequences [15,16]. MRI measurements of diaphragmatic excursion (DE) can be performed at reproducible locations and accurate timings during the respiratory cycle. It is considered to offer a precise measurement of DE [17]. Ultrasound measurements of DE have not yet been compared with those obtained using another technique. Since critical care physicians are using exclusively US to evaluate diaphragmatic function in their unstable patients, we were interested in verifying whether MRI (operator-independent) and US measurements of DE are interchangeable. The primary objective of this study was to assess the agreement between DE measurements obtained using dynamic MRI with those obtained using ultrasound in patients. The secondary objectives were: 1) to compare the diagnostic performance of MRI and US for recognizing DD, and 2) to compare US and MRI-measured DE thresholds to define DD.

## Material and methodsMaterials and methods

### Setting

We conducted a prospective single-center study between May 7, 2019 and January 7, 2020, in which all consecutive outpatients referred for a dynamic thoracic MRI were enrolled (N = 45). Indications for obtaining a dynamic MRI included the exploration of known (n = 14) or suspected acquired DD (n = 4), or other thoracic disorders (n = 27).

## Ultrasound and MRI assessment of diaphragmatic excursion

Each patient underwent sequentially US and dynamic MRI explorations of the diaphragm. The order for the procedures was randomly assigned. The DE of each hemi-diaphragm was measured using US and MRI by trained operators blinded to patient's clinical information and to each other during, both, quiet spontaneous breathing (SB) and forced inspiration (FI) consecutively. Patients were always studied in the supine position.

The US assessment was performed using a Compact Xtreme CX50 (Philips, Einthoven, Netherlands) with a low-frequency probe (2.5–3.5 MHz). Before starting the procedure, the patient was trained to perform forced inspiration maneuvers. All patients were equipped with a three-branch ECG that also allowed acquiring the respiratory cycle trace from the changes in thoracic impedance. The technique for US measurement of DE has been described previously [17]. For the measurement of DE during quiet SB, the final value was calculated as the mean of 3 consecutive respiratory cycles. The DE during FI was the maximum value obtained during this maneuver. Representative examples of ultrasound studies for DE measurements are presented in Fig 1.

MRI studies were obtained on a 3T Discovery 750 w MR scanner (GE Healthcare, Waukesha, WI, USA) using AIR coils technology. The image acquisition was synchronized to the respiratory cycles of the patient. Dynamic MRI sequences were performed with a balanced steady-state free precession sequence (FIESTA) in sagittal (6 slices) and coronal views (3 slices) with the following parameters: repetition time (msec)/echo time (msec): 3.198/1.236; $224 \times 256$ matrix; bandwidth: 488 Hz; flip angle: 50°; and temporal resolution: 60 (coronal) or 35 (sagittal) phases (1 image/sec). The acquisition time for the complete diaphragmatic exploration was around 30 minutes. The DE was measured from end-inspiration to end-expiration for each hemi-diaphragm during FI and SB. As for US, each value of excursion during SB was

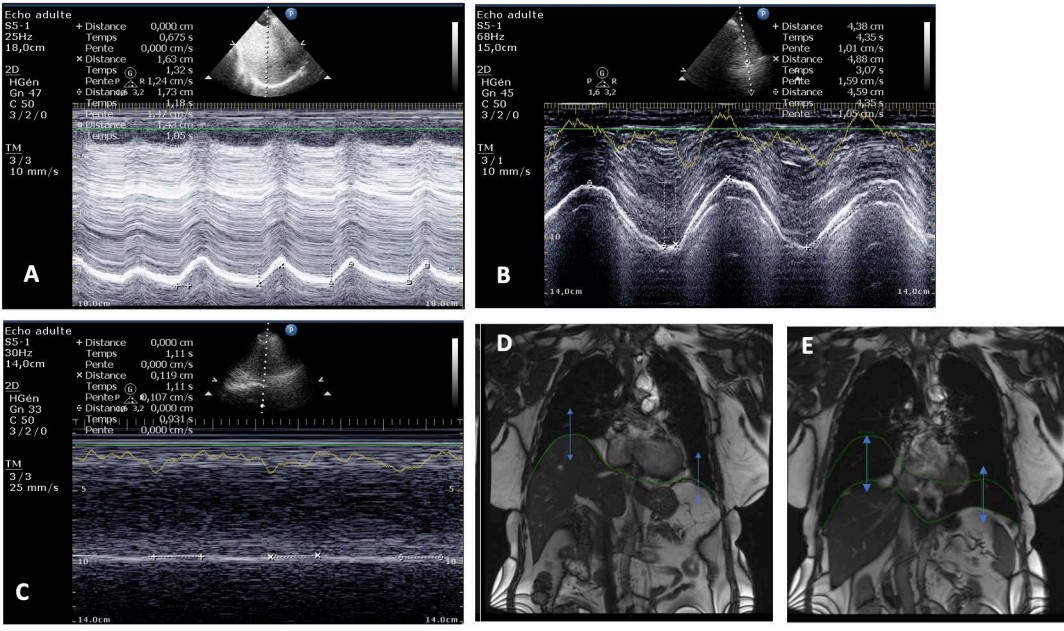

**Fig 1. Representative examples of ultrasound DE measurements obtained during spontaneous breathing (Panels A and C) and during forced inspiration (B).** Patient with normal diaphragmatic function: panels A and B. Patient with diaphragmatic paralysis: panel C. Example of a diaphragmatic MRI study in coronal view used for the measurement of DE. Excursion (double arrow) is defined as the difference of diaphragmatic position between full inspiration (D) and full expiration (E).

calculated as the average of 3 respiratory cycles, and as the single best value during FI. We then compared the excursion obtained from the MRI coronal view at the vertex of each hemi-diaphragm with the paired US measurement. The coronal view in the middle of the hemi-diaphragm was used for the measurement of DE. Representative examples of images obtained with dynamic MRI are presented in Fig 1.

## Data collection

Patient characteristics included medical and surgical history, risk factors for DD, respiratory functional status (modified Medical Research Council mMRC score) [18], and the results of the respiratory function tests. These data were extracted from the patients' medical records after the study was completed.

## Definition of diaphragmatic dysfunction

Threshold values for DE defining DD have only been published for US measurements. The patient was recognized as having DD if the diaphragmatic excursion measured using US was 10 mm or less for men, or 9 mm or less for women during regular quiet SB, and less than 47 mm for men and 37 mm for women during FI, as proposed by Boussuges *et al.*

Diaphragmatic paralysis was defined by a paradoxical displacement or an absence of motion (DE ≤ 0 mm) of the hemi-diaphragm [1,19].

## Primary and secondary endpoints

The primary endpoint was the agreement between US and MRI measurements of DE for each hemi-diaphragm. The secondary endpoints of the study included: 1) comparison of DE threshold values attesting for DD using US and dynamic MRI and 2) comparison of US and MRI performance in recognizing DD.

## Ethics

This study was approved by an independent institutional review board (CPP Rennes Ouest V, registration number: 2019-A00005-52). All patients provided written informed consent before enrollment.

## Statistical analysis

Since this was a descriptive exploratory study, estimating a sample size to provide adequate power to our analysis was not appropriate. Categorical data are presented as numbers and percentages, and values are expressed as mean±SD or as median and Inter-Quartile Ranges (IQR), as appropriate. We used Bland and Altman representations to describe the agreement between US and MRI for the measurement of DE, and we calculated the bias and the limits of agreement with 95% confidence intervals (95% CI). Student's t-test tested the null hypothesis that the bias was zero. To evaluate the correlation between the measures of DD we calculated Lin's concordance correlation coefficient (CCC) [20] Using the validated US DE cut-offs defining DD, we constructed 2000 Receiver Operating Characteristics (ROC) curves for different thresholds of DE using dynamic MRI. The threshold for DE obtained using MRI was selected to maximize sensitivity and specificity as measured in Youden's J statistic. The discriminative performance of MRI was evaluated by calculating the area under curve (AUC) of the ROC curve. Ninety five percent confidence intervals for the AUC were calculated via the bootstrap with 2000 iterations using the pROC package [21]. All statistical analyzes were performed using the R statistical software version 4.0.0 (http://www.R-project.org, the R

Foundation for Statistical Computing, Vienna, Austria). All tests were bilateral and a value of p < 0.05 was considered statistically significant.

## Results

### Population characteristics

Forty-five patients were referred for a dynamic thoracic MRI assessment over the eight-month study period. The flow of patients in the study is presented in Fig 2. Four patients had to be excluded from the study due to claustrophobia preventing completion of the MRI study (n = 2), or because of unsuspected diaphragmatic rupture (n = 2). In addition, 3 patients could not perform forced inspiration and 1 patient had irregular SB precluding the acquisition of the corresponding sequences. Forty patients underwent the complete sequence of data acquisition during SB, and 38 during FI, for both US and MRI. Patient demographic data (age, sex, weight, height and BMI), prior surgical history at risk of diaphragmatic dysfunction, risk factors for DD, respiratory functional status (modified Medical Research Council mMRC score) [18], and the results of the respiratory function tests are presented in Table 1.

### Diaphragmatic excursions measured using US and MRI for each hemi-diaphragm

The values for left and right DE during spontaneous breathing and forced inspiration obtained using US and MRI are presented in Table 2.

### Agreement between ultrasound and MRI for the measurement of diaphragmatic excursion

Bland and Altman graphs are presented in Fig 3. During SB, the mean bias for DE measurement was −3.8 mm, 95% CI [−7.1; −0.6] for the left hemi-diaphragm, and 1.0 mm, 95% CI [−3.5; 5.5] for the right hemi-diaphragm. Limits of agreement (millimeters) were [−25; 17] on the left side, and [−28; 30] on the right side.

The systematic biases for right DE measurement during SB (1 mm, p = 0.86) and during FI (−5.7 mm, p = 0.08) were not significantly different from 0 mm. On the contrary, on the left side, SB breathing bias (−3.8 mm, p = 0.04) and FI bias (−5.1 mm, p = 0.007) were both significantly different from 0 mm.

Lin's CCC obtained during SB was 0.45, 95% CI [0.20; 0.64] for the right side, and 0.52, 95% CI [0.27; 0.70] for the left side. For the right hemi-diaphragm, the CCC during FI was 0.77, 95% CI [0.62; 0.87] for the right side and 0.77, 95% CI [0.61; 0.87] for the left side.

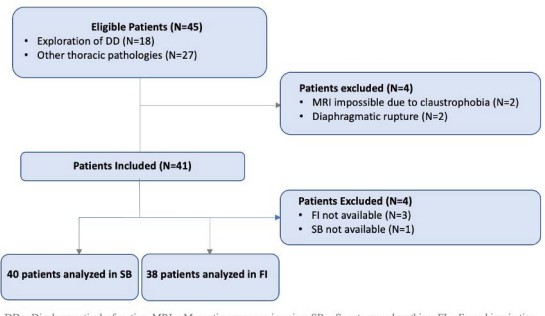

**Fig 2. Flow of participants through the study.**

**Table 1. Patient's characteristics.**

| Variables | Patients (N = 41) |
|---|---|
| Age, mean ± SD, y | 58 ± 24 |
| Male Sex | 25 (61%) |
| Female Sex | 16 (39%) |
| Weight, kg | 75 ± 15 |
| Height, cm | 169 ± 7 |
| BMI, kg/m² | 26 ± 6 |
| Overweight (BMI > 25) | 26 (63%) |
| Obese (BMI > 30) | 2 (5%) |
| COPD | 5 (12%) |
| Diabetes | 2 (5%) |
| Chronic Heart Failure | 2 (5%) |
| Current smoker | 9 (22%) |
| Former smoker | 8 (19%) |
| Never smoke | 24 (59%) |
| Pack-years, mean ± SD | 31 ± 19 |
| Alcohol consumption > 3 standard units/day | 7 (17%) |
| Diaphragmatic Dysfunction | 18 (44%) |
| Diaphragmatic plication | 3 (17%) |
| Phrenic paralysis and diaphragmatic plication | 2 (11%) |
| Post-surgical diaphragmatic paralysis | 5 (28%) |
| Diaphragmatic hernia | 4 (22%) |
| Ascended diaphragmatic cupola on chest X-ray | 4 (22%) |
| Prior surgery with high risk of DD (thoracic, upper abdominal, or cardiac)[*] | 23 (56%) |
| thoracic | 18 (44%) |
| cardiac | 4 (10%) |
| upper abdominal | 8 (19%) |
| None | 18 (44%) |
| Functional respiratory tests available | 18 (50%) |
| Obstructive ventilatory disorder - FEV/FVC < 70%- | 4 (22%) |
| Restrictive ventilatory disorder -TLC < 80% | 6 (33%) |
| Mixed ventilatory disorder -obstructive restrictive- | 1 (6%) |
| Normal FRT | 7 (39%) |
| 6-minute walk test: mean ± SD, m | 386 ± 96 |
| Patients with an FRT measuring the difference in vital capacity between standing and supine positions | 13 (32.5%) |
| Negative: difference < 15% | 9 (69%) |
| Positive: difference > 20% | 4 (31%) |

[*]Some patients had more than one type of surgery at risk for DD, and more than one diaphragmatic disorder. BMI = Body Mass Index,

DD = diaphragmatic dysfunction, FEV = Forced Expiratory Volume, FVC = Forced Vital Capacity, FRT = Functional Respiratory Tests, TLC = Total Lung Capacity.

## Sensitivity, specificity, threshold, and probability of diaphragmatic dysfunction

The thresholds for diaphragmatic excursion measured using MRI that offered the optimal diagnostic performances to identify DD are presented in Table 3.

**Table 2. Diaphragmatic Excursion (mm) measured using US and MRI.**

| Breathing condition | US | | MRI | |
|---|---|---|---|---|
| | Left | Right | Left | Right |
| SB | 17,3 ±11 | 16,3 ±11 | 13 ±11 | 18 ±17 |
| FI | 43,6 ±22 | 49,2 ±27 | 34,9 ±21 | 42,7 ±32 |

Values are mean±SD. DE = Diaphragmatic Excursion, SB = Spontaneous Breathing, FI = Forced Inspiration.

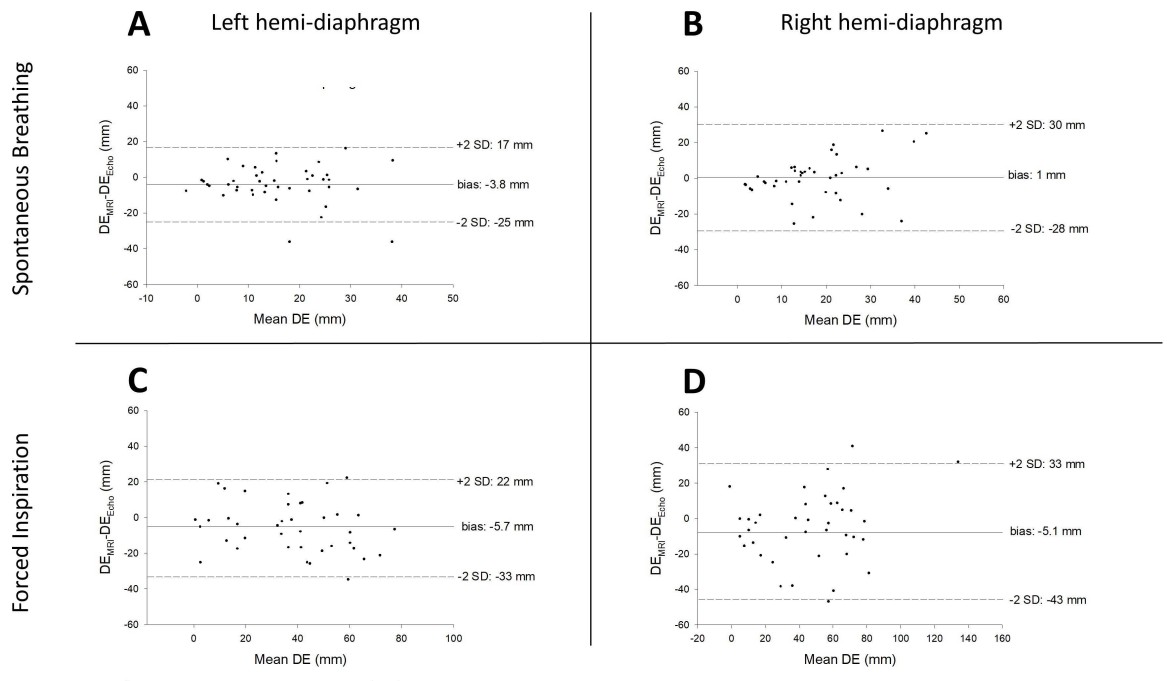

DE = Diaphragmatic Excursion, SD = Standard Deviation

**Fig 3. Bland and Altman graphs showing the agreement between US and MRI measurements of diaphragmatic excursion obtained during quiet spontaneous breathing (panels A and B) and during forced inspiration (panels C and D), for the left (panels A and C) and right (panels B and D) hemi-diaphragms.**

**Table 3. Sensibility, Specificity and AUC for the prediction of DD during SB and FI.**

| Breathing condition | Threshold MRI (mm) | Sensitivity (%) | Specificity (%) | AUC |
|---|---|---|---|---|
| SB | 11 | 77,7 | 77,4 | 0,86 |
| FI | 38 | 77,4 | 75,5 | 0,87 |

AUC = Area Under Curve, SB = Spontaneous Breathing, FI = Forces Inspiration.

The ROC curves depicting the probability of detecting DD using MRI in relation to the probability of DD predicted by US are presented in Fig 4.

The logistic regression curve and the calibration curve of the logistic model used to compare ultrasound and MRI SB and FI are presented in the S1 and S2 Figs.

## Discussion

In this exploratory study comparing US and MRI for the measurement of diaphragmatic excursion, we observed that the agreement between the two techniques was not very tight, suggesting that operators tracked the displacement of different parts of the diaphragmatic dome when using ultrasound or MRI. However, both measures were well correlated, especially during FI. The thresholds defining DD for MRI were similar to those published for US. These thresholds yielded good sensitivity and specificity to recognize diaphragmatic dysfunction.

### Comparison of DE measurements between ultrasound and MRI

There were differences between the DE values measured using US and dynamic MRI for both hemi-diaphragms, during quiet SB and FI. The most likely explanation is that US and MRI measurements were probably not acquired from the exact same location on the diaphragmatic dome. In addition, despite the care taken in carrying out the US measurements, it is possible that we did not track the exact same point between end-expiration and end-inspiration, creating variability in the measurement of excursion. Conversely, MRI measurements were acquired on precise anatomical landmarks on the diaphragmatic dome. There was no systematic bias for the measurements obtained on the right side during spontaneous breathing. This is not surprising since this is the easiest DE measurement to acquire: the subcostal view offers a very good US visualization of the right hemi-diaphragm with perfect alignment with the displacement. On the other hand, measurements obtained from the left side or from the right side during FI were all overestimated when acquired with US. Several factors may explain this lack of agreement. During spontaneous breathing, measurements of the left DE were acquired

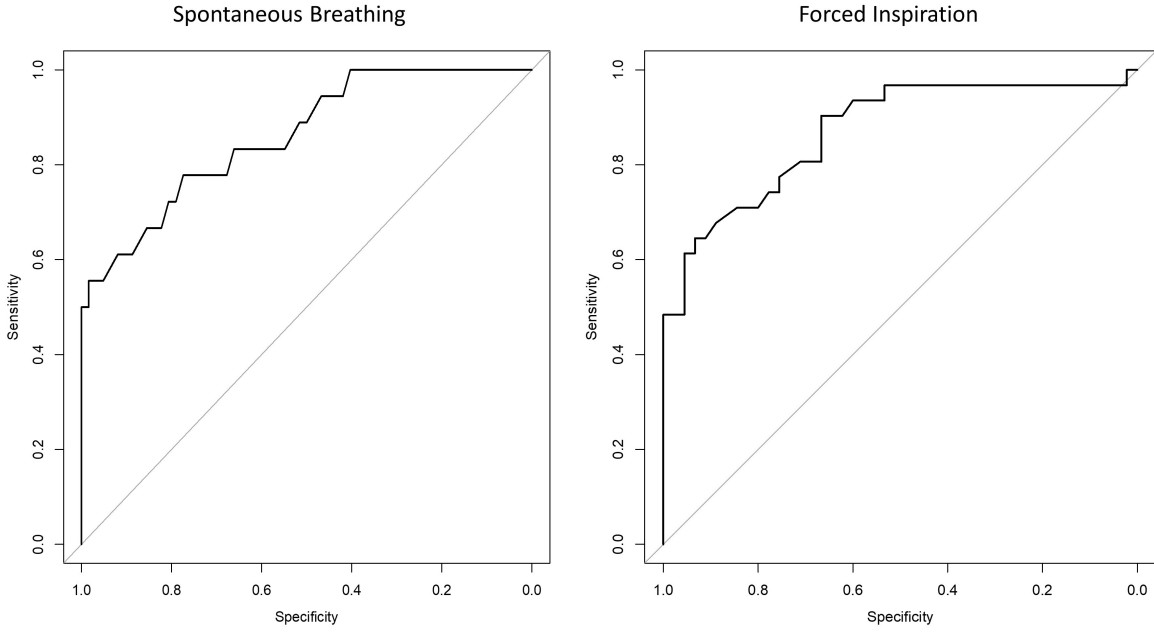

**Fig 4. ROC curves depicting the probability of detecting DD using MRI in relation to the probability of DD predicted by US.**

from the transthoracic approach, which requires an angle correction using anatomical M-Mode because the alignment with the displacement is not possible from this vantage point. It was previously demonstrated that failing to correct for this lack of alignment might result in DE overestimation using M-Mode [22]. It is therefore possible that operators did not properly correct for the misalignment, resulting in some degree of DE overestimation using US. This overestimation was always more pronounced during FI because it is very difficult to ensure that the same portion of the diaphragmatic dome is tracked properly during deep breathing maneuvers using US.

The combination of suboptimal ultrasonic alignment and anatomical positioning explains the imperfect agreement between MRI and US measurements of DE. Interestingly, concordance correlation coefficients between the two techniques were better during FI than SB, suggesting that this maneuver reduced the discrepancy between measurements. However, FI is rarely possible to obtain in critically ill patients and this population is most often studied during quiet SB [5]. Although MRI exploration of the diaphragm usually relies on FI maneuvers, we were interested in studying DE measurements obtained during SB, since this is how critically ill patients are most often explored [3].

## Diagnostic performances

Up to now, there are no published values of DE measured using MRI that allow to qualify the diaphragm as normal or dysfunctional. Only US have been used in cohort of subjects to establish thresholds of normal values for DE during quiet SB and FI [1]. We have therefore used the US measurements of DE to characterize the diaphragmatic function of our subjects. In order to determine the MRI threshold values for DE defining dysfunction, we used a bootstrapping technique, *i.e.,*: a method of statistical inference allowing to create new samples of measures (resampling) from the data of the study population characterized using US. This analysis yielded MRI thresholds of 11 mm for SB, and 38 mm for FI, values that appear to be very similar to the thresholds of Boussuges *et al.* for US (9–10 mm and 37–47 mm, respectively) [1]. In addition, we found that the sensitivity and the specificity of these MRI thresholds for the diagnosis of DE were high, with 77.7% of sensitivity and 77.4% of specificity during SB, and 77.4% and 75.5% during FI. The AUC of the ROC curves were also good (0.86 and 0.87), attesting for the high predictive value of these thresholds. Although the agreement between DE measurements obtained using US or MRI was not perfect, the diagnostic performance of the two techniques appeared comparable. This is of course reassuring for intensive care physicians who must rely on US to assess the diaphragmatic function of their patients, since MRI is usually not feasible in the context of critical illness.

## Limits and strengths of the study

To our knowledge, this study is the first that compared US and MRI for the diagnosis of DD in a sample of patients undergoing dynamic MRI of the diaphragm. The cohort included patients with a balanced proportion of male and female with pathological and healthy diaphragms. We were able to determine threshold values for DE measured using MRI corresponding to diaphragmatic dysfunction according to US. Only 18 patients had functional respiratory tests (FRT), and therefore we did not attempt to study correlations between DE and FRT parameters.

This study was conducted in outpatients referred for a dynamic MRI of the diaphragm, and not in critically ill patients. We were not able to implement a gold standard technique to qualify diaphragmatic dysfunction, due to the invasiveness and discomfort of these methods.

## Conclusions

US and MRI provided different results for diaphragmatic excursion, probably because the measurements were not obtained exactly at the same location. Nevertheless, diagnostic performances of both techniques to assess diaphragmatic dysfunction appear comparable, a finding that reinforces the confidence that ICU physicians can have in US for this purpose.

## Suppporting information

**S1 Fig. Logisitic Regression curve of the DD in US compared to MRI in spontaneous breathing and forced inspiration.**
(TIF)

**S2 Fig. Callibration curve of the logistic model used to compare ultrasound and MRI during Spontaneous breathing and forced inspiration.**
(TIF)

## Acknowledgments

This work has been presented by the authords as an abstract at the ATS (American Thoracic Society) in 2021.

## Author contributions

**Conceptualization:** Clara Delplancke, Etienne Charpentier, Hélène Nougué, Françoise Le Pimpec-Barthes, Bernard Cholley, Matthieu Daniel.

**Data curation:** Clara Delplancke, François Grolleau, Bernard Cholley, Matthieu Daniel.

**Formal analysis:** Clara Delplancke, Etienne Charpentier, François Grolleau, Bernard Cholley, Matthieu Daniel.

**Funding acquisition:** Clara Delplancke, Matthieu Daniel.

**Investigation:** Clara Delplancke, Etienne Charpentier, Anne Hernigou, Bernard Cholley, Matthieu Daniel.

**Methodology:** Clara Delplancke, Etienne Charpentier, François Grolleau, Hélène Nougué, Bernard Cholley, Matthieu Daniel.

**Project administration:** Clara Delplancke, Etienne Charpentier, Hélène Nougué, Matthieu Daniel.

**Resources:** Clara Delplancke, Hélène Nougué, Matthieu Daniel.

**Software:** Clara Delplancke, Matthieu Daniel.

**Supervision:** Clara Delplancke, Bernard Cholley, Matthieu Daniel.

**Validation:** Clara Delplancke, Etienne Charpentier, Bernard Cholley, Matthieu Daniel.

**Visualization:** Clara Delplancke, Matthieu Daniel.

**Writing – original draft:** Clara Delplancke, Bernard Cholley, Matthieu Daniel.

**Writing – review & editing:** Clara Delplancke, Hélène Nougué, Françoise Le Pimpec-Barthes, Bernard Cholley, Matthieu Daniel.

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
