## [Decision Letter · Decision Letter 0]

25 Nov 2024

PONE-D-24-27511Comparison of ultrasound and dynamic MRI for the measurement of diaphragmatic excursion: a prospective single-center studyPLOS ONE

Dear Dr. Delplancke,

Thank you for submitting your manuscript to PLOS ONE. After careful consideration, we feel that it has merit but does not fully meet PLOS ONE’s publication criteria as it currently stands. Therefore, we invite you to submit a revised version of the manuscript that addresses the points raised during the review process.

**ACADEMIC EDITOR: **

We look forward to receiving your revised manuscript.

Kind regards,

Hidetaka Hamasaki

Academic Editor

PLOS ONE

Additional Editor Comments (if provided):

Reviewers' comments:

Reviewer's Responses to Questions

**Comments to the Author**

1. Is the manuscript technically sound, and do the data support the conclusions?

Reviewer #1: Yes

Reviewer #2: Partly

2. Has the statistical analysis been performed appropriately and rigorously? 

Reviewer #1: Yes

Reviewer #2: No

3. Have the authors made all data underlying the findings in their manuscript fully available?

Reviewer #1: Yes

Reviewer #2: No

4. Is the manuscript presented in an intelligible fashion and written in standard English?

Reviewer #1: Yes

Reviewer #2: No

5. Review Comments to the Author

Reviewer #1: Dear Authors;

This is a good paper. It only lacks good comparison with RECENT literature such as the following SUGGESTIVE papers:

https://publications.ersnet.org/content/erj/58/5/2100137.abstract

https://link.springer.com/article/10.1186/s12890-021-01441-6

https://journals.lww.com/cptj/abstract/2024/10000/differences_in_diaphragmatic_and_chest_wall.4.aspx?context=latestarticles

https://link.springer.com/article/10.1186/s12903-023-03558-y

https://www.tandfonline.com/doi/full/10.2147/IJGM.S478136

https://link.springer.com/article/10.1186/s43168-024-00315-9

https://link.springer.com/article/10.1007/s40477-021-00570-2

https://link.springer.com/article/10.1007/s00247-022-05430-7

The above links are ONLY examples, you may consider all or part of them but the MRI vs. Ultrasound are crucial (they need to be considered), or you may add more by visiting google scholar searching recent works in the field. You need to illustrate how your results are different from recent attempts. I expect that this should be written in the text somewhere in the Discussion section of your paper. Subsequently, you need to explain how that this paper has not been addressed by other scholars (i.e. ONLY what are the differences to claim this strong statement?)

Finally, It is very important to explain how the same title of this paper was found in Google Scholar, the link is below:

https://www.atsjournals.org/doi/pdf/10.1164/ajrccm-conference.2021.203.1_MeetingAbstracts.A4669.

Minor concern: In Figure 1, the sum of patients in the first block (Upper block) should be 45 not 44.

Reviewer #2: Thank you so much to the authors for allowing me to review their research.

Abstract:

The methodology should be clearly summarized to indicate the research approach, setting and data collection methods.

Introduction:

The introduction seems to be well written, but the authors need to ensure that all information should be references accordingly if it is not general knowledge. Line 62 - 63 page 2 should include a reference.

Materials and: Methods

This section requires careful attention as there are several areas that are not clearly presented. There is no clear presentation of the research methodology and approach that was used. I suggest including the geographical area where the study was performed even if the actual institution is not included. There is lack of detail regarding the setting in terms of the level of health care, modalities available etc. Include a heading to cover the population and sampling clearly and in detail because the numbers (N and n) are not adding up correctly. N which should represent the total population included in the study is not correctly presented in several areas in the study.

Ensure that all your supplementary files are linked and referenced correctly in text. Data collection should also be clearly outlined as the information provided is vague.

Results:

The information in Table 1 does not seem entirely balanced or internally consistent.

There is a mismatch in sample size the table indicates n=41, fig 1 suggests a total n = 45 while a sum of subcategories (18 + 26_ totals n = 44. The gender breakdown specifies 25 males (61%) but does not provide the number of females, which is necessary to complete the demographic profile. Some subcategories fail to add up to the total sample size n=14. For example, in Diaphragmatic plication, the percentages total 78% (n = 3 + n = 2 + n = 5 + n = 4). For Smoking history, the sum of current smoker (n = 9), former smoker (n = 8), and never smoked (n = 24) equals 41, which aligns with N. However, this pattern should be consistent across all variables. There are also incomplete subcategory definitions in your table. Some variables (e.g., Prior surgery with high risk of DD) provide subcategories (e.g., thoracic, cardiac, abdominal), but the sum of these does not align with the total count. Example: Subcategories under "Prior surgery" add up to 30, which exceeds the total N = 41. there is lack of clarity in percentages and units. Percentages in the table should be double-checked: Example: For "BMI > 25", 26 patients (63%) are consistent with N = 41, but for other rows (e.g., "Obese"), percentages and counts are less clear. Also have a look at the Functional respiratory test (FTR) Subcategories: • The breakdown under "Patients with a difference between supine and standing positions" is problematic: Total = 13 (32.5%), with a split of 9 (69%) negative and 4 (31%) positive. However, these percentages should align with n = 13, not the total N.

Discussion and conclusion:

Align after completion of amendments.

6. PLOS authors have the option to publish the peer review history of their article (what does this mean? ). If published, this will include your full peer review and any attached files.

**Do you want your identity to be public for this peer review?** For information about this choice, including consent withdrawal, please see our Privacy Policy .

Reviewer #1: **Yes: ** Abdel-Razzak Al-Hinnawi

Reviewer #2: No

---

## [Author Response · Author response to Decision Letter 1]

8 Jan 2025

We sincerely thank the reviewers for their constructive feedback. Please find below our point-by-point responses to the comments. All the changes made in the manuscript appear in red.

Responses to Reviewer #1:

This is a good paper. It only lacks good comparison with RECENT literature such as the following SUGGESTIVE papers

https://publications.ersnet.org/content/erj/58/5/2100137.abstract

https://link.springer.com/article/10.1186/s12890-021-01441-6

https://journals.lww.com/cptj/abstract/2024/10000/differences_in_diaphragmatic_and_chest_wall.4.aspx?context=latestarticles

https://link.springer.com/article/10.1186/s12903-023-03558-y

https://www.tandfonline.com/doi/full/10.2147/IJGM.S478136

https://link.springer.com/article/10.1186/s43168-024-00315-9

https://link.springer.com/article/10.1007/s40477-021-00570-2

https://link.springer.com/article/10.1007/s00247-022-05430-7

The above links are ONLY examples, you may consider all or part of them but the MRI vs. Ultrasound are crucial (they need to be considered), or you may add more by visiting google scholar searching recent works in the field. You need to illustrate how your results are different from recent attempts. I expect that this should be written in the text somewhere in the Discussion section of your paper. Subsequently, you need to explain how that this paper has not been addressed by other scholars (i.e. ONLY what are the differences to claim this strong statement?)

We thank the reviewer for his appreciation and valuable suggestions. We fully agree that the inclusion of recent articles is essential. Accordingly, we have now cited Keyes S. et al. (page 3, line 73) to highlight the value of ultrasound in measuring diaphragmatic excursion and Harlaar L. et al. (page 3, line 79) to underscore the relevance of MRI for the same purpose.

We have conducted new bibliographic research on the comparison between ultrasound (US) and MRI for the measurement of diaphragmatic excursion. We can confirm that, to the best of our knowledge, no original research directly addressing the comparison between these two techniques is available to date. Existing articles are either general reviews or comparisons involving other methods.

Finally, It is very important to explain how the same title of this paper was found in Google Scholar, the link is below:

https://www.atsjournals.org/doi/pdf/10.1164/ajrccm-conference.2021.203.1_MeetingAbstracts.A4669.

This work was previously presented by the authors as an abstract at the ATS meeting in 2021. This information has now been included in the Acknowledgements section.

Minor concern: In Figure 1, the sum of patients in the first block (Upper block) should be 45 not 44.

We thank the reviewer for his careful reading of the figures. This mistake has now been corrected in Figure 2.

Responses to Reviewer #2:

Abstract:

The methodology should be clearly summarized to indicate the research approach, setting and data collection methods.

Following the reviewer’s suggestion, we have included some methodological details in the abstract, keeping in mind that we are limited with the number of words allowed.

Introduction:

The introduction seems to be well written, but the authors need to ensure that all information should be references accordingly if it is not general knowledge. Line 62 - 63 page 2 should include a reference.

To comply with the reviewer’s advice, we have added the reference of a general review on diaphragmatic dysfunction published in the New England Journal of medicine (Page 3, line 60 and 65).

Ref#5: McCool FD, Tzelepis GE. Dysfunction of the diaphragm. N Engl J Med. 2012;366: 932–942. doi:10.1056/NEJMra1007236).

Materials and Methods

This section requires careful attention as there are several areas that are not clearly presented. There is no clear presentation of the research methodology and approach that was used. I suggest including the geographical area where the study was performed even if the actual institution is not included. There is lack of detail regarding the setting in terms of the level of health care, modalities available etc.

To clarify the setting of the Institution where the research was conducted, we have added the following sentence page 2, line 40-42: “Prospective single-center study conducted at a tertiary university hospital, where extensive expertise in diaphragmatic US and MRI was available. Consecutive outpatients referred for a dynamic thoracic MRI were enrolled.” In addition, page 4, line 91-92, we now specify the type of institution and the geographical area: “We conducted a prospective single-center study between May 7, 2019 and January 7, 2020, at a tertiary university hospital in France”.

Include a heading to cover the population and sampling clearly and in detail because the numbers (N and n) are not adding up correctly. N which should represent the total population included in the study is not correctly presented in several areas in the study.

We apologize because there was a typo in the “N” of Figure 2 (44 instead of 45). This has been corrected in figure 2, and the N=45 is now mentioned in the “setting section (page 4, line 93) to ensure clarity. We have also added the following sentence in the “setting” section of the Material and Methods (page 4, line 93): “Indications for obtaining a dynamic MRI included the exploration of known (n=14) or suspected acquired DD (n=4), or other thoracic disorders (n=27).”

Ensure that all your supplementary files are linked and referenced correctly in text.

We apologize for this omission. Supplementary Figures 1 and 2 are now quoted in the manuscript page 10 and line 249-251: “The logistic regression curve and the calibration curve of the logistic model used to compare ultrasound and MRI during spontaneous SB and FI are presented in the supplementary figures 1 and 2”.

Data collection should also be clearly outlined as the information provided is vague.

We describe more precisely the data collected (page 7, line 190-194): “Patient demographic data (age, sex, weight, height and BMI), prior surgical history at risk of diaphragmatic dysfunction, risk factors for DD, respiratory functional status (modified Medical Research Council mMRC score) [17], and the results of the respiratory function tests are presented in Table 1. “

Results:

The information in Table 1 does not seem entirely balanced or internally consistent.

There is a mismatch in sample size the table indicates n=41, fig 1 suggests a total n = 45 while a sum of subcategories (18 + 26_ totals n = 44).

We appreciate the careful review of the results and the identification of this mistake (18 + 27 = 45). This error has now been corrected in Figure 2, with the total updated to N=45 instead of 44.

The gender breakdown specifies 25 males (61%) but does not provide the number of females, which is necessary to complete the demographic profile.

We have added the number of females in Table 1:

Some subcategories fail to add up to the total sample size n=14. For example, in Diaphragmatic plication, the percentages total 78% (n = 3 + n = 2 + n = 5 + n = 4).

This discrepancy results from the fact that some patients had more than 1 diaphragmatic disorder or more than 1 surgery at risk for DD, therefore it doesn’t strictly add up. To clarify, we have modified the caption of Table 1: “Some patients had more than one type of surgery at risk for DD or more than one diaphragmatic disorder”.

There are also incomplete subcategory definitions in your table. Some variables (e.g., Prior surgery with high risk of DD) provide subcategories (e.g., thoracic, cardiac, abdominal), but the sum of these does not align with the total count. Example: Subcategories under "Prior surgery" add up to 30, which exceeds the total N = 41.

Again, this is because some patients underwent more than one surgery at risk for the diaphragm, i.e.: thoracic + abdominal. We hope that the new caption of Table 1 clarifies this.

There is lack of clarity in percentages and units. Percentages in the table should be double-checked: Example: For "BMI > 25", 26 patients (63%) are consistent with N = 41, but for other rows (e.g., "Obese"), percentages and counts are less clear.

For clarity and simplicity, the percentages have been rounded to the nearest whole number. For instance, 2 patients were classified as obese: 2/41 = 4.87%, which was rounded up to 5%. We have thoroughly double-checked all these percentages and we confirm their accuracy.

Also have a look at the Functional respiratory test (FTR) Subcategories: • The breakdown under "Patients with a difference between supine and standing positions" is problematic: Total = 13 (32.5%), with a split of 9 (69%) negative and 4 (31%) positive. However, these percentages should align with n = 13, not the total N.

We apologize for the lack of clarity surrounding this point. In fact, 13 patients underwent specific “diaphragmatic FRT” measuring the difference in vital capacity between supine and standing positions. Among those, 9/13 (69%) had less than 15% of difference between the 2 positions (confirming the absence of DD), whereas 4/13 (31%) had a difference considered positive (>20%) confirming the existence of DD. We have clarified this directly in Table 1: “patients with an FRT measuring the difference in vital capacity between standing and supine positions”.

On behalf of the writing team, we hope this document thoroughly addresses all the insightful points raised by the reviewers, responding to each one point by point.

---

## [Decision Letter · Decision Letter 1]

21 Jan 2025

Comparison of ultrasound and dynamic MRI for the measurement of diaphragmatic excursion: a prospective single-center study

PONE-D-24-27511R1

Dear Dr. Delplancke,

We’re pleased to inform you that your manuscript has been judged scientifically suitable for publication and will be formally accepted for publication once it meets all outstanding technical requirements.

Kind regards,

Hidetaka Hamasaki

Academic Editor

PLOS ONE

Additional Editor Comments (optional):

Reviewers' comments:

Reviewer's Responses to Questions

**Comments to the Author**

1. If the authors have adequately addressed your comments raised in a previous round of review and you feel that this manuscript is now acceptable for publication, you may indicate that here to bypass the “Comments to the Author” section, enter your conflict of interest statement in the “Confidential to Editor” section, and submit your "Accept" recommendation.

Reviewer #1: All comments have been addressed

Reviewer #2: All comments have been addressed

2. Is the manuscript technically sound, and do the data support the conclusions?

Reviewer #1: Yes

Reviewer #2: Yes

3. Has the statistical analysis been performed appropriately and rigorously? 

Reviewer #1: Yes

Reviewer #2: Yes

4. Have the authors made all data underlying the findings in their manuscript fully available?

Reviewer #1: Yes

Reviewer #2: Yes

5. Is the manuscript presented in an intelligible fashion and written in standard English?

Reviewer #1: Yes

Reviewer #2: Yes

6. Review Comments to the Author

Reviewer #1: (No Response)

Reviewer #2: Introduction

Line 62 - 63 page 2 should include a reference. - Reference still not included “This dysfunction results from a neuromuscular disorder that 65 can involve the central nervous system, a peripheral neuropathy, or a myopathy.” REF

Methods and materials

Setting

page 2, line 40-42: “Prospective single-center study conducted at a tertiary university hospital, where extensive expertise in diaphragmatic US and MRI was available. Consecutive outpatients referred for a dynamic thoracic MRI were enrolled.” - This information should be removed from the abstract and added under the sub-heading ‘setting’

page 10 and line 249-251: - Figure 1 and 2 are provided as supplementary files. I suggest labelling them as e.g. "Figure S1," "Figure S2," and so on, to distinguish them from figures that are typically placed in the main manuscript. Ensure that the caption for each supplementary figure should include a detailed caption explaining its content and relevance.

This discrepancy results from the fact that some patients had more than 1 diaphragmatic disorder or more than 1 surgery at risk for DD, therefore it doesn’t strictly add up. To clarify, we have modified the caption of Table 1: “Some patients had more than one type of surgery at risk for DD or more than one diaphragmatic disorder” - This should be clearly outlined in the text. If the authors have done so - thanks. If not please add it.

Again, this is because some patients underwent more than one surgery at risk for the diaphragm, i.e.: thoracic + abdominal. We hope that the new caption of Table 1 clarifies this. - This should be clearly outlined in the text. If the authors have done so - thanks. If not please add it.

We apologize for the lack of clarity surrounding this point. In fact, 13 patients underwent specific “diaphragmatic FRT” measuring the difference in vital capacity between supine and standing positions. Among those, 9/13 (69%) had less than 15% of difference between the 2 positions (confirming the absence of DD), whereas 4/13 (31%) had a difference considered positive (>20%) confirming the existence of DD. We have clarified this directly in Table 1: “patients with an FRT measuring the difference in vital capacity between standing and supine positions”. - All this information should be outlined clearly in the manuscript to ensure that the results are clearly outlined to transform raw data into meaningful insights, thus helping the manuscript to communicate its scientific value effectively. This action underscores the credibility of the research and fosters a deeper understanding of its implications.

7. PLOS authors have the option to publish the peer review history of their article (what does this mean? ). If published, this will include your full peer review and any attached files.

**Do you want your identity to be public for this peer review?** For information about this choice, including consent withdrawal, please see our Privacy Policy .

Reviewer #1: **Yes: ** Abdel-Razzak Al-Hinnawi

Reviewer #2: No

---

## [Editor Report · Acceptance letter]

PONE-D-24-27511R1

PLOS ONE

Dear Dr. Delplancke,

I'm pleased to inform you that your manuscript has been deemed suitable for publication in PLOS ONE. Congratulations! Your manuscript is now being handed over to our production team.

Kind regards,

on behalf of

Dr. Hidetaka Hamasaki

Academic Editor

PLOS ONE